# Gamma Irradiation and Male *Glossina austeni* Mating Performance

**DOI:** 10.3390/insects11080522

**Published:** 2020-08-11

**Authors:** Chantel J. de Beer, Percy Moyaba, Solomon N. B. Boikanyo, Daphney Majatladi, Gert J. Venter, Marc J. B. Vreysen

**Affiliations:** 1Insect Pest Control Laboratory, Joint FAO/IAEA Division of Nuclear Techniques in Food and Agriculture, Seibersdorf 1400, Austria; M.Vreysen@iaea.org; 2Epidemiology, Vectors and Parasites, Agricultural Research Council-Onderstepoort Veterinary Research (ARC-OVR), Onderstepoort 0110, South Africa; MoyabaP@arc.agric.za (P.M.); BoikanyoS@arc.agric.za (S.N.B.B.); MajatladiD@arc.agric.za (D.M.); venterjgert@gmail.com (G.J.V.); 3Department of Veterinary Tropical Diseases, University of Pretoria, Onderstepoort 0110, South Africa

**Keywords:** sterile insect technique, radiation sensitivity, relative mating index

## Abstract

**Simple Summary:**

African animal trypanosomosis, also known as nagana, is a neglected disease in South Africa, transmitted by tsetse flies. It has been proposed to manage this disease through the eradication of *Glossina austeni*, one of the vectors of this disease in South Africa. The strategy would be an integration of various control tactics and the release of sterilised colonised males (sterile insect technique) to eliminate relic pockets. The irradiated colonised males must be able to compete with wild males. In preparation of the technique, the mating performance of colony-reared male flies was assessed. Factors that can influence male mating performance, such as radiation dose, and the development stage that is exposed to radiation were evaluated in the laboratory and under semi-field conditions. Radiation doses of 80 Gy and 100 Gy induced 97–99% sterility in colony females that mated with colony males treated as adults or pupae. Walk-in field cage assessments indicated that a dose of up to 100 Gy did not adversely affect the mating performance of males irradiated as adults or late stage pupae. This study indicated that the colonized *G. austeni* males irradiated as adults or late stage pupae will be suited for the sterile insect technique.

**Abstract:**

An area-wide integrated pest management (AW-IPM) strategy with a sterile insect technique (SIT) component has been proposed for the management of African animal trypanosomosis (AAT) in South Africa. In preparation for the SIT, the mating performance of colony reared *Glossina austeni* males under influencing factors such as radiation dose and the development stage that is exposed to radiation, was assessed under laboratory and semi-field conditions. The radiation sensitivity of *G. austeni* colonized 37 years ago when treated as adults and late-stage pupae was determined. Radiation doses of 80 Gy and 100 Gy induced 97–99% sterility in colony females that mated with colony males treated as adults or pupae. Males irradiated either as adults or pupae with a radiation dose of 100 Gy showed similar insemination ability and survival as untreated males. Walk-in field cage assessments indicated that a dose of up to 100 Gy did not adversely affect the mating performance of males irradiated as adults or late stage pupae. Males irradiated as adults formed mating pairs faster than fertile males and males irradiated as pupae. The mating performance studies indicated that the colonized *G. austeni* males irradiated as adults or late stage pupae will still be suited for SIT.

## 1. Introduction

Although the sterilizing effect of radiation on insects was already known in the 1930s [1,2], only 20 years later in the 1950s the potential of this technique for the management of insect pests was realized [3,4]. The sterile insect technique (SIT) necessitates the radiation sterilization of male insects obtained from mass-rearing of the target species [5] followed by the release of these males in sufficient numbers to outcompete their wild counterparts [6,7]. Since the SIT is species-specific it can be considered as an environment-friendly control method [8]. The mating of the sterile males with wild fertile females results in no progeny, which leads to a reduction and in some cases to local eradication of the target insect pest population [7]. The natural slow reproduction rate in tsetse flies will contribute to the success of the SIT.

The feasibility of the SIT for the eradication of entire tsetse populations has been demonstrated on several occasions [9,10,11]. The first campaign that incorporated the use of radiation-sterilized adults with aerial spraying of insecticides to suppress the wild fly population was implemented in Tanzania in the 1970s [12]. This was followed by an effort in the 1980s in Burkina Faso, where suppression methods such as insecticide impregnated targets were integrated with the release of sterile males. This resulted in the local eradication of populations of *Glossina morsitans submorsitans*, *Glossina palpalis gambiensis* and *Glossina tachinoides* [13]. Similarly, *Glossina palpalis palpalis* was locally eradicated from 1500 km^2^ of agro-pastoral land in the Jos area of Nigeria [14]. Although the programs in Burkina Faso and Nigeria managed to locally eradicate the tsetse population, the control strategies did not adhere to area-wide integrated pest management (AW-IPM) principles, and the pest free status of these areas was subsequently lost due to reinvasion [6].

The most successful AW-IPM program that had a SIT component to eradicate tsetse flies to date was implemented on Unguja Island, Zanzibar, in the 1990s [9]. Suppression of *Glossina austeni* by means of insecticide-treated screens and pour-on application of insecticides on cattle was started in 1988. The suppression was followed by the aerial release of 8.5 million sterile male flies between August 1994 and December 1997. The last wild *G. austeni* fly was sampled in September 1996, and the last infection with *Trypanosoma vivax* recorded in 1998 [15]. At the time of writing, Unguja Island is still free of tsetse flies and trypanosomosis.

In 2005, the Government of Senegal embarked on a similar AW-IPM project to eradicate a *G. palpalis gambiensis* population in the Niayes area, integrating the SIT with suppression methods like insecticide impregnated screens, pour-on applications and ground spraying of insecticides in the hot spot areas [16,17,18]. The sterile males for the Niayes program were mass-reared in three production centers in Burkina Faso, Slovakia and Austria, and irradiated male pupae (110 Gy) were transported under chilled conditions by courier to Dakar in Senegal [19,20] where they emerged after arrival [20,21,22] and prepared for release [19,20]. The flies were released as adults, after having received two blood meals, using either carton release boxes or an automated chilled adult release device mounted on a gyrocopter [23,24].

There are positive and negative considerations when deciding to irradiate adult flies or pupae [25]. Adult pupae are easier to transport, more robust to handle and larger volumes of pupae can be irradiated simultaneously [19].

The mating competitiveness of males irradiated either as adults or pupae needs to be considered in deciding which stage to irradiate. The use of field cages to assess mating performance and competitiveness of pest insects has gained in importance during the last decade. These cages were originally designed for testing the mating behavior of fruit flies (Drosophilidae) [26,27], but their use has been expanded to tsetse flies [21,22,28,29,30,31,32], mosquitoes [33,34] and Lepidoptera [35,36,37,38]. Walk-in field cages have proved to be good substitutes for field studies that are more complex and expensive with many parameters that cannot be controlled.

Two tsetse species, *Glossina brevipalpis* and *G. austeni*, are present in an area of 16,000 km^2^ in the north eastern part of the KwaZulu-Natal Province in South Africa [39,40], and both populations extend into southern Mozambique and *G. austeni* into eSwatini [41,42,43]. Although *G. austeni* is considered to be the more competent vector for the transmission of *Trypanosoma* parasites, it has been shown that both species can transmit *Trypanosoma congolense* and *Trypanosoma vivax* in South Africa [44,45,46,47]. An AW-IPM strategy that includes a SIT component was proposed to establish a tsetse fly free South Africa [48]. The successful implementation of an AW-IPM program with a SIT component depends on a number of prerequisites [49] of which the biological quality and sexual competitiveness of the mass-reared sterile males are amongst the more important ones [50].

To be successful, the released sterile males must be able to compete with local wild males [51]. Releasing low quality sterile males will necessitate higher release rates, require more funding that might prolong the duration of the program and potential failure [52]. A recent study indicated that treating late stage *G. brevipalpis* pupae with 40 Gy induced 97% sterility when mated with untreated females and 99% sterility when irradiated with 80 Gy as adults [25]. High insemination rates (91.9–100%) were observed irrespective whether the males were irradiated as adults or pupae. This study further showed that males irradiated as adults with 80 Gy successfully competed with colonized fertile males for colony females under semi-field conditions [25], indicating that the *G. brevipalpis* colony at the Agricultural Research Council-Onderstepoort Veterinary Institute (ARC-OVI) can provide sterile males suitable for a release program [25].

The successful colonization of *G. austeni* [52,53,54,55,56,57,58,59] enabled studies on the radiation sterilization of adult males. A radiation dose of up to 120 Gy had no visible effect on the mating behavior, insemination rate, sperm motility and competitiveness in *G. austeni* [60,61]. During multiple mating experiments with untreated females and untreated and irradiated males, females used predominantly sperm from the first mating for fertilization, irrespective of whether the male was sterile or fertile [60,61].

The successful eradication of *G. austeni* from the Island of Unguja [9] demonstrated the feasibility of using the SIT to eradicate a geographical isolated population of this species and that colonized *G. austeni* can be used. However, only limited information is available on radiation sensitivity of pupae, the subsequent competitiveness of the emerging adult sterile males and on their mating behavior in general. Observations on their mating behavior in the laboratory and field cages may identify potential problems with the field-released flies. In view of the importance of this species in the transmission of *Trypanosoma* parasites in South Africa and the limited knowledge on the effect of radiation of immatures on the mating performance, the radiation sensitivity and mating performance of *G. austeni* males treated as adults and pupae were assessed.

## 2. Materials and Methods

### 2.1. Experimental Flies

Adults and pupae of *G. austeni* used in this study were derived from the laboratory colony maintained at the ARC-OVI since 2002. This colony was established at the ARC-OVI in Pretoria, South Africa using seed material from the Vector and Vector-Borne Diseases Research Institute, Tanga (formerly known as the Tsetse and Trypanosomiasis Research Institute (TTRI)). The original *G. austeni* colony was established at the TTRI in September 1982 from pupae collected in the Jozani Forest, Unguja Island of Zanzibar, and this colony was also used for the mass-production of the flies used in the Unguja AW-IPM program [9,62]. Experimental flies and pupae were kept under standard colony conditions (23–24 °C, 75–80% RH and indirect lighting) [63,64]. Flies were kept in standard holding cages (Ø 20 cm) and fed daily on defibrinated bovine blood using an artificial in vitro membrane feeding system.

### 2.2. Evaluation of Radiation Sensitivity of Males

Adult *G. austeni* males and pupae were given a radiation dose in air of either 40 Gy, 80 Gy, 100 Gy, 120 Gy or 140 Gy in a ^137^Cs source (Gammacell 40 S/N50) at a dose rate of 0.69 Gy/min. A group of fertile non-irradiated fertile males were used as a control. Adult males were irradiated on day four post-emergence. The pupae were collected from the colony in 24 h intervals to synchronize adult emergence and irradiated at three specific times, i.e., on days 36 (group 1), 34 (group 2) or 32 (group 3) post-larviposition. Adult emergence rate of the irradiated pupae was recorded.

To determine their reproductive success, six-day-old males from all treatments (irradiated pupae/adults and fertile males) were mated with three-day-old virgin females at a 1:2 male (*N* = 15): female (*N* = 30) ratio. Males and females were kept together for four days to mate, where after they were separated by chilling at 4 °C and maintained at a density of 30 females and 30 males per standard holding cage (Ø 20 cm). All treatments were repeated three to four times.

Male and female mortality was monitored daily and female pupae production recorded for 60 days. Fecundity was defined as the number of pupae produced per mature female day [61]. Mature female days were calculated for each treatment by adding the number of flies alive each day, starting on day 18 after emergence until the end of the experiment on day 60 [61]. All pupae produced were mechanically sorted into one of five class sizes (A–E). The sorter was calibrated according to the FAO/IAEA standard, i.e., a range between 2.3 (A) and 3.0 mm (E). The weight (mg) of the different pupal size classes were A (≤16), B (16 ≤ 19), C (19 ≤ 21), D (21 ≤ 23) and E (≥23).

Abortions of eggs and immature larval stages were monitored daily. After 60 days all surviving females were dissected to determine their reproductive status, insemination rate and spermatecal fill [64,65,66,67]. The content of the uterus was examined for eggs or larvae. The spermathecae were removed and the fill microscopically scored as empty (0), quarter full (0.25), half (0.5), three quarters (0.75) or full (1) [68]. Male mortality was monitored until all the males had died.

### 2.3. Male Mating Performance

The effect of irradiation on male mating performance of *G. austeni* was assessed in two walk-in field cage experiments, i.e., the effect of the radiation dose and the effect of irradiation during different development stages (adults and pupae). Adult males were irradiated in air four days after emergence with a dose of 80 Gy or 100 Gy and the pupae were irradiated three days before emergence with a dose of 100 Gy. For both experiments, the irradiated males competed with non-irradiated fertile males for matings with virgin females. The experiment was replicated twelve times.

The mating performance of *G. austeni* was assessed in walk-in field cages under “near-natural” conditions [29,69]. The cylindrical field cages (Ø 2.9 m × 2.0 m) were made of cream polyester netting with a flat floor and ceiling. Black nylon strips, connecting the panels of polyester netting, encircle the top and bottom of the cage where the ceiling and floor meet the sides of the cage. A 1.5 m potted weeping boer-bean tree (*Schotia brachypetala*) was placed in the middle of the cage. A zip, also in a black nylon strip, from top to bottom sealed the entrance of the cage. The field cages were deployed in a small irrigated forest, at the ARC-OVI in Pretoria South Africa, as described previously [25]. The field cage experiments were carried out in the afternoon from 12:00 to 15:00.

Throughout the experiment, temperature and relative humidity were recorded every 10 min using a DS1923-F5# Hygrochron iButton data logger (Fairbridge Technologies, Johannesburg, South Africa). Light intensity at the top and the bottom of the cage and at tree level was recorded every 15 min using a Major Tech MT940 light meter (Major Teck, Johannesburg, South Africa).

At the start of the field cage experiments, 30 three-day-old virgin females were released in the middle of the cage 5 min before 90 males (two treatment groups and one untreated group, 30 males per group) were released, giving a male to female ratio of 3:1. Male flies were marked with a dot of polymer paint on the notum 24 h before being released with a different color to differentiate the various male groups [29]. Flies were not fed on the day of the field cage experiment to limit inactivity due to blood meal digestion.

The observer remained inside the cage for the 3 h duration of each experiment and movements were kept to a minimum. The time of mating was recorded to determine mating latency. The mating pairs were collected individually into small vials, and duration of the mating recorded. Although no direct adverse effect on mating behavior was seen when the pairs were collected, its potential influence on mating behavior cannot be ruled out. To minimize this effect mating pairs were collected using the same protocol in all experiments. These mating pairs were not replaced.

Females that did not mate after three hours were immobilized at −5 °C and dissected the following day to confirm virginity. All males remaining in the cages at the end of the experiments were returned to the colony. The mated females and males were transferred to individual holding cages and kept for 60 days and fed daily on defibrinated bovine blood.

The females were monitored daily for survival and production similar to that of the radiation sensitivity experiment. Females were dissected after 60 days to determine reproductive status, insemination rate and spermathecae fill [63,64].

The propensity of mating (PM), relative mating index (RMI) and relative mating performance (RMP) were the mating indices used to compare the mating performance of the males of the various treatments. Propensity of mating (PM) was defined as the overall proportion of released females that had mated. Relative mating index (RMI) was defined as the number of pairs of one treatment group as a proportion of the total number of matings. Relative mating performance (RMP) was defined as the difference between the number of matings of two treatments of males as a proportion of the total number of matings [29]. In addition, the mating latency time, mating duration, insemination rate and the spermathecae fill of each mated female were determined.

### 2.4. Statistical Analysis

Data were analyzed using the statistical software GraphPad Instat [70]. Proportional differences in adult emergence rates were determined with chi-square (χ^2^) analysis with the Yate’s continuity correction. Linear and multiple regression analyses were carried out on fecundity as well as male survival in relation to the radiation dose.

Differences in the overall proportions of peak mating activity were analyzed with Chi-square (χ^2^) analysis with the Yate’s continuity correction. The *p* value was two-sided and a relative risk, p1–p2 was determined. Additionally, an unpaired test was used to differentiate between the average mating latency, mating duration and spermathecae fill. 

A one-way analysis of variance (ANOVA) was used to differentiate between the relative mating index, average mating latency, mating duration and spermathecae fill. Where the data passed the normality test, standard (parametric) methods were used and the Tukey’s test was applied. If the data was not normally distributed the nonparametric Kruskal–Wallis test was used. All tests were done at the 5% significance level.

## 3. Results

### 3.1. Male Radiation Sensitivity

#### 3.1.1. Adult Emergence Timing and Rate

*Glossina austeni* females from group 1 pupae started to emerge on day 33–34 post-larviposition with a peak emergence on day 37. Male emergence in this group commenced on day 36–37, peaked on day 38 and stopped on day 40 post-larviposition. A similar pattern was observed in group 2 and 3 pupae with females that started to emerge on day 33–35 and on day 31–35 post-larviposition, respectively. Female emergence from group 3 pupae peaked between day 36 and 38 and for group 2 pupae on day 36 post-larviposition. The male flies from group 2 and 3 pupae started to emerge between day 36 and 38 and on day 36 to 37, respectively. Adult male emergence of both groups peaked on day 39 post-larviposition and no more emergence was seen in both groups on day 41 post-larviposition.

The adult emergence rate of the group 2 pupae (95.6%) was significantly higher than the total emergence rate of group 1 pupae (91.1%; *p* < 0.01, *χ*^2^ = 8.31, d.f. = 1) as well as of group 3 pupae (88.1%; *p* < 0.01, *χ*^2^ = 21.90, d.f. = 1). There was no significant difference in the emergence rates of group 2 and 3 pupae. In all treatment groups pupae emerged in equal female to male ratios.

#### 3.1.2. Female and Male Survival

Of the 2730 *G. austeni* females (30 for each treatment replicate) at the onset of the experiments, 2449 (89.7%) survived to day 18 post-emergence. The average survival rate to day 18 of females that mated with the males irradiated as adults ranged from 85.00% ± 6.93% (100 Gy) to 90.83% ± 4.19% (fertile males; Table 1). For females that mated with males irradiated as pupa from pupa group 1, 2 and 3 the average survival rate ranged from 80.83% ± 12.87% (fertile males) to 95.0% ± 1.92% (80 Gy), 83.33% ± 18.56% (fertile males) to 97.78% ± 1.92% (120 Gy) and 82.50% ± 24.55% (fertile males) to 94.16% ± 5.00% (140 Gy), respectively (Table 1). The survival rate for females decreased at the end of the experiment (day 60) to an average of 67.91% ± 6.68% for the females that mated with the males irradiated as adults, 75.83% ± 6.35%, 72.96% ± 9.95% and 82.42% ± 7.26% for the females that mated with males irradiated as pupa from pupae group 1, 2 and 3 respectively Table 1.

The average survival rate at day 30 of males irradiated as adults ranged from 33.33% ± 8.61% (120 Gy) to 58.33% ± 8.39% (80 Gy; Table 1). For males irradiated as pupae the average survival rate ranged from 23.09% ± 10.18% (pupae irradiated 34 days post-larviposition) to 76.67% ± 3.84% (pupae irradiated 32 days post-larviposition; Group 3; Table 1). A negative linear correlation was found between the average lifespan for the males irradiated as adults (*r^2^* = 0.9.2, *p* = 0.01), however this correlation was not observed for the males irradiated as pupae (group 1, *r^2^* = 0.71, *p* = 0.07; group 2, *r^2^* = 0.66, *p* = 0.09 and group 3, *r^2^* = 0.61, *p* = 0.11).

#### 3.1.3. Fecundity of Females that Mated with Irradiated Males

As the radiation dose increased, the number of pupae produced per mature female (fecundity) that mated with irradiated males decreased proportionally (Table 1). A negative linear regression was found between fecundity of untreated females and radiation dose administered to their male mates as adults (*r^2^* = 0.81, *p* < 0.01) and as group 1 (*r^2^* = 0.72, *p* < 0.01), group 2 (*r^2^* = 0.73, *p* < 0.01) and group 3 (*r^2^* = 0.71, *p* < 0.01), pupae. A dose of 40 Gy induced 61.5% sterility in females that had mated with males irradiated as adults. In females mated with males emerging from group 1 and 2 pupae, the sterility was 61.6%, whereas it was 77.3% for females mated with males from group 3 pupae. Higher doses of 80 Gy and 100 Gy induced 97–99% sterility in females that mated with males treated as adults or pupae. The pupal production (Table 1) relative to the controls (fertile males) for females mated with males irradiated as adults with a dose of 80 Gy was 2.6% and for males treated as pupae it was 2.2% in group 1, 3.1% in group 2 and 1.8% in group 3 pupae. Using a dose of 100 Gy the pupal production (Table 1) relative to the controls (fertile males) was 1.3%, 2.5% and 2.2% for females mated with males irradiated as pupae of group 1, 2 and 3, respectively. No pupae were produced by females that mated with males treated as adults with a dose of 100 Gy.

The number of eggs aborted during the 60-day experimental period was lower in females that mated with non-irradiated males than in females mated with any of the males in the experimental groups (Table 1). For all treatment groups most of the pupae produced were in or above the pupal size class C, i.e., between 19 and 21 mg (Table 1). The male to female ratio of flies that emerged from pupae produced by females mated with irradiated males were equally distributed (1:1 ratio; Table 1).

The average insemination rate of females (Table 2) that were mated with males irradiated as adults (98.1% ± 2.96%) was higher than that in the fertile males group (93.9% ± 5.44%) and similar high insemination rates, independent of dose, were observed when the males were irradiated as pupae. The high insemination rates, independent of dose, indicated that treatment of both adult and pupae males did not affect their ability to transfer sperm to the females (Table 2). There was a marked difference in the reproductive status of females that mated with non-irradiated males compared with treated males as indicated by dissection results (Table 2). The uterus of females mated with irradiated males, irrespective of if they were irradiated as adults or pupae, were either empty due to abortions or contained a recently ovulated egg, almost none of these females had any viable larvae in utero (Table 2). In contrast, the uterus of females that had mated with non-irradiated males as well as females mated with males irradiated with a dose of 40 Gy dose contained either a recently ovulated egg or a viable instar larva, and fewer females were found with an empty uterus due to an abortion (Table 2).

### 3.2. Male Mating Performance

#### 3.2.1. Environmental Conditions in Field Cage

Field cage experiments to assess mating performance of males irradiated with different irradiation doses and irradiated at different live stages (as adults or pupae) were carried out in September 2014 and November 2016, respectively. September is the last month of the dry season in Pretoria, and the mean temperature and relative humidity recorded during the field cage experiment were 29.50 ± 1.52 °C and 21.70% ± 6.46%. The mean temperature and relative humidity recorded for November was 26.10 ± 4.12 °C and 47.90% ± 17.60%. In September, the light intensity ranged from 910.30 ± 149.31 Lx at the top of the field cage to 653.50 ± 125.02 Lx at the bottom. This was due to the seasonal change in leaf cover of the trees, being less in the dry season. In November the light intensity, was 524.50 ± 170.31 Lx at the top to 547.00 ± 130.53 Lx at the bottom of the cages. The light intensity ranged from an average of 805.40 ± 90.74 Lx and 638.50 ± 154.77 Lx measured at the tree inside the cage in September and November, respectively.

#### 3.2.2. Mating Performance of Males

A propensity of mating (PM; overall proportion of released females that mated) of 0.63 and 0.41 was obtained in the evaluation of different irradiation doses and for the live stage evaluation, respectively (Table 3). The average relative mating index (RMI) was not significantly different (*p* = 0.91) for the untreated males (0.33 ± 0.13) compared to either the males that were irradiated as adults with a dose of 100 Gy (0.32 ± 0.12) or 80 Gy (0.35 ± 0.16; Table 3). Similarly, there was also no significant difference in the RMI (*p* = 0.91) for males irradiated as adults (0.24 ± 0.14), or as pupae (0.36 ± 0.14) or for untreated males (0.39 ± 05).

The relative mating performance (RMP; the difference between the numbers of matings of sterile to fertile males as a proportion of the total number of matings) of males irradiated with 100 Gy, as adults were −0.03, indicating that fertile males were more successful in mating with the females. The RMP of 80 Gy-treated *G. austeni* males as compared to untreated ones was 0.01 (in favor of treated males). In the evaluation of irradiated males as pupae or adults, the RMP of sterile males irradiated as adults versus fertile males was −0.03 in favor of the fertile males and −0.14 for males irradiated as pupae versus fertile flies, again in favor of fertile males.

Males irradiated with a dose of 80 Gy (79.00 ± 0.49 min) as well as a 100 Gy (69.90 ± 0.04 min) mated on average sooner than the untreated males (81.00 ± 0.04 min). Males irradiated as adults formed mating pairs faster (71.70 ± 0.36 min) than the fertile males (85.20 ± 0.04 min) and the males irradiated as pupae (84.00 ± 0.06 min). In both evaluations the fertile males mated the longest (dose evaluation: 155.10 ± 0.04 min; live stage evaluation: 126.10 ± 0.04 min; Table 3). The insemination rate for both the dose rate as well as live stage evaluation were above 0.96 and the mean spermathecal value above 0.58 ± 0.31, which is an indication that irradiation did not influence the males’ ability to transfer sperm (Table 3).

## 4. Discussion

The success of an AW-IPM programs that include a SIT component will depend on the ability of the released sterile males to compete with wild males for mating opportunities with wild females. Therefore, an assessment of the mating competitiveness of the produced, released insects is an essential prerequisite for the success of SIT programs [49,50,71]. This also means that the selection of an appropriate radiation dose to sterilize the males that will be released in these programs will be crucial. A dose below the optimal will result in insects that are not sufficiently sterile, and a too high dose may result in insects with impaired quality that are not competitive with wild flies [71,72,73]. Factors such as the developmental stage of the insect, its age and the atmosphere used during irradiation can influence the required dose and the level of sterility achieved [71,72].

A study on the effect of gamma irradiation on *G. austeni* pupae indicated that a dose of 70 Gy of gamma rays stimulated emergence of *G. austeni* pupae [74]. Late stage pupae were less radiation sensitive then younger pupae but feeding frequency, meal size, fat body development, longevity and thoracic flight muscle development were not affected by irradiation (80–200 Gy) in the pupal stage [75]. Our data are in agreement with previous studies [61,67], in that the amount of dominant lethal mutations induced in the sperm of *G. austeni* when exposed to radiation as expressed by induced sterility in the female mates, increased with increasing dose. In the present study a dose of 80 Gy or 100 Gy induced a minimum of 97% sterility in *G. austeni* females that mated with males treated either as adults or pupae. These irradiation doses are lower than that of 120 Gy previously proposed [76,77], a dose that was also used in the program that successfully eradicated *G. austeni* from Unguja Island [9]. Another study used doses of 50 Gy, 60 Gy and 70 Gy to produce partially sterile *G. austeni* males and showed that these were sufficient to obtain dominant lethality in 70–90% of their sperm, which is comparable to our data [78].

The present data indicates that *G. austeni* pupae are more radiosensitive than adults, as was observed for *G. palpalis palpalis* [65]. These authors found that the rate of pupae production relative to the control for a dose of 80 Gy was 7.10 for irradiated adults and 4.08 for irradiated pupae. Dissection results revealed a clear abortion pattern in *G. palpalis palpalis* females that mated with males treated with 80 Gy or higher. The uterus of females that mated with irradiated males contained either an egg in embryonic arrest or was empty as a result of abortion of the embryo or the larva. It was suggested that this observation could be used to monitor the impact of sterile male releases on a natural tsetse fly population [66]. The imbalance between uterus content and the follicle next in the ovulation sequence [67] was indeed used successfully to assess with a high level of accuracy the rate of induced sterility in the wild female *G. austeni* population during the eradication campaign on Unguja Island [9] and against *G. palpalis gambiensis* in the Niayes of Senegal [18]. The natural abortion rate of the population needs to be determined first for comparison with abortion rates during the sterile male release program. Similarly, *G. brevipalpis* pupae were more radiosensitive than adults with 93% and 97% sterility induced in females that were mated with males irradiated with 40 Gy as adults and late stage pupae, respectively. In addition, the longevity of the males was more compromised when pupae were irradiated as compared with adults [25].

The reduction in average longevity of irradiated males compared to that of untreated males is a manifestation of the somatic damage caused by irradiation [9]. The negative linear regression between dose and longevity for *G. austeni* was only observed when adults were irradiated. In contrast, average longevity of adult males irradiated as pupae on day 36 post-larviposition was significantly increased. This radiation-induced increase in average lifespan of males was also documented for *G. morsitans* pupae irradiated in air [79] and nitrogen [80] and for *G. brevipalpis* males irradiated with low doses (10–40 Gy) as adults [67]. Many reasons for this increase in average lifespan have been proposed and it is most likely that a range of radiation induced repair mechanisms are involved [81].

The results of the present study indicate that *G. austeni* can be treated as late pupae (up to day 32 post-larviposition, which is about seven days before emergence) with a dose of 100 Gy for use in an AW-IPM program with an SIT component. In addition to the small differences in the quality of sterile males when irradiated as adults or pupae, other logistical requirements, e.g., distance of mass-rearing facility and radiation source from release site, may need to be taken into consideration when selecting the most efficient irradiation protocol. Although it is known that exposure to radiation may affect the biological quality of the produced released insects [82], our data indicated that irradiation of adults or late stage pupae with up to a 100 Gy did not affect the ability of sterilized males to compete with fertile males under semi-field conditions. This is in accordance with field cage evaluations of *G. morsitans* and *G. pallidipes* that also showed that the competitiveness of irradiated males did not differ from that of untreated males [29,83]. Furthermore, their findings of competitiveness were based on pupal production, however, in this study as well as others observing individual flies in walk-in field cages provided more accurate information on male competitiveness [29,83].

In our study, an average propensity of mating between 41% and 63% was observed, which, although being lower than that found in similar field cage experiments with *G. fuscipes fuscipes* and *G. palpalis palpalis* [28], was indicative of the adequate environmental conditions for the evaluations. In addition, this relatively high propensity of mating indicated that the observer’s presence had only minimal interference on the mating behavior of the flies. It needs, however, to be pointed out that the field cage experiments were conducted in Pretoria, which has a different climate than the tsetse-fly infested area in North-Eastern KwaZulu-Natal. The different environmental conditions might influence the circadian rhythm, the activity patterns of the flies and the propensity of mating.

In the release program on Unguja Island, quality control of the sterile *G. austeni* males included laboratory as well as field evaluations. The laboratory evaluations included the assessment of longevity and mating success of 120 Gy-irradiated *G. austeni* males, the survival of males kept at different cage densities, transportation simulation tests and assessment of accessary glad development in virgin males. There was a significant reduction in male lifespan when males were irradiated however, all matings with irradiated males resulted in normal insemination and 100% induced sterility. Male densities in release cages did not influence longevity. There was a negative effect on male survival and mating performance with relation to time between transportation and release and type of marking material used. The field evaluations included the assessment of sterile male quality during transport, after release and the survival of released males in the forest habitat. In the field cage evaluations, the average longevity of irradiated and unirradiated 1–2, 4 and 8 day old male *G. austeni* was similar [84].

Whereas in the programs in Burkina Faso, Nigeria, Unguja Island and Senegal the sterile male insects were released as adults, in the program against *Glossina morsitans morsitans* in the Tanga area of Tanzania in the 1970s, the sterile males were released as pupae from fixed release stations. The emerging males were consequently teneral and had to find immediately a blood meal to build up energy reserves [85]. A drawback of this method was that males were exposed to potential predation before reaching sexual maturity and could potentially transmit the *Trypanosoma* parasites. Despite this, the program was successful and releasing male pupae at a density of 135 pupae/km^2^ resulted in a sterile male wild male overflooding ratio of 1.2:1, which, despite being low, maintained the indigenous wild fly population at the 80–95% reduction level obtained after the initial insecticide application [86].

*Glossina brevipalpis* seems to be more radiosensitive than *G. austeni* [25,67], which indicated that *G. brevipalpis* can be treated either as adult or late pupa (up to seven days before emergence) with a dose of 80 Gy and *G. austeni* with a dose of 100 Gy for use in AW-IPM programs with a SIT component.

## 5. Conclusions

For the SIT, mass-reared and released sterile males must be able to compete successfully with their wild counterparts for mating with wild females. *Glossina austeni* males irradiated with a dose of 100 Gy, either as adults or late stage pupae, showed a similar insemination ability and survival as untreated males. Additionally, walk-in field cage assessments indicated that a dose of up to 100 Gy did not adversely affect the mating performance of males irradiated as adults or late stage pupae. Males irradiated as adults formed mating pairs faster than fertile males and males irradiated as pupae. The fertile males mated the longest, however the insemination rate was still above 0.96 for all matings. The data obtained for *G. austeni* strengthens the findings of previous studies that indicate that this species can be used successfully in AW-IPM programs with an SIT component.

## Figures and Tables

**Table 1 insects-11-00522-t001:** Production of *Glossina austeni* females mated with males irradiated as adults or pupae 3, 5 or 8 days before emergence.

Radiation Dose (Gy)	No. of Replicates	Pupae Emergence	% Male Survival at Day 30 (Avg. ± SD)	No. of Mature Females at Day 18	% Female Survival) at		No. of Pupae Produced	% Pupal Size Classes in Mg					Fecundity	% Emergence/% Females
					Day 18 (Avg. ± SD)	Day 60 (Avg. ± SD)		A <16	B 16 ≤ 19	C 19 ≤ 21	D 21 ≤ 23	E <23		
	Irradiated Adult Males													
Fertile Males	4	-	60.00 ± 17.21	109	90.83 ± 4.19	65.00 ± 6.38	234	7.70	7.30	12.80	21.80	50.40	2.15	93.20/55.10
40	4	-	56.67 ± 13.88	106	88.33 ± 6.93	69.17 ± 12.58	90	7.80	11.10	12.20	16.70	52.20	0.85	92.20/53.00
80	4	-	58.33 ± 8.39	107	89.17 ± 5.00	71.67 ± 14.00	6	33.30	0	16.70	16.70	33.30	0.06	83.30/60.00
100	4	-	53.33 ± 23.09	102	85.00 ± 6.93	55.83 ± 5.69	0	0	0	0	0	0	0	
120	4	-	33.33 ± 8.61	109	90.83 ± 3.19	74.17 ± 8.77	1	0	0	0	0	0	0.01	(1/1)/(0/1)
140	4	-	36.67 ± 5.44	105	87.50 ± 4.19	71.67 ± 6.94	1	0	0	0	0	0	0.01	(0/1)
	Pupae Irradiated 36 Days Post-Larviposition (Group 1)													
Fertile Males	4	82.1	83.33 ± 18.35	97	89.17 ± 17.29	80.0 ± 20.18	226	1.80	4	13.30	23	58	2.33	89.4/59.4
40	4	86.4	41.66 ± 6.38	110	91.67 ± 6.94	77.50 ± 6.93	70	1.40	2.90	10	14.30	71.40	0.64	87.1/54.4
80	4	85.4	35.00 ± 10.00	114	95.00 ± 1.92	84.17 ± 5.00	5	0	0	0	40	60	0.04	4/5/1/5
100	4	98.1	43.33 ± 31.03	110	91.07 ± 8.81	75.00 ± 15.52	3	0	0	0	33.30	66.70	0.03	2/3/1/3
120	4	96.1	41.667 ± 4.87	111	92.50 ± 10.67	72.50 ± 17.08	2	0	0	0	0	100	0.02	2/2/2/2
140	4	89.3	48.33 ± 23.33	107	80.83 ± 12.87	65.83 ± 15.00	4	0	50	25	0	25	0.04	4/4/1/4
	Pupae Irradiated 34 Days Post-Larviposition (Group 2)													
Fertile Males	3	80.5	33.33 ± 3.84	75	83.33 ± 18.56	60.00 ± 26.03	159	10.10	6.30	28.90	18.20	36.50	2.12	57.90/45.7
40	3	100	37.12 ± 7.69	78	86.67 ± 3.33	68.89 ± 22.19	61	3.30	4.90	34.40	29.50	27.90	0.78	80.30/63.30
80	3	85.4	53.33 ± 13.87	78	86.67 ± 8.82	74.44 ± 12.62	5	0	0	20	60	20	0.06	(3/5)/(2/5)
100	3	98.1	60.00 ± 13.33	87	96.67 ± 0.00	66.67 ± 24.03	4	25	0	50	0	25	0.05	(2/4)/1/4
120	3	96.1	51.11 ± 16.94	88	97.78 ± 1.92	87.78 ± 10.72	0	0	0	0	0	0	0	
140	3	89.3	23.09 ± 10.18	112	91.11 ± 10.18	80.00 ± 6.67	0	0	0	0	0	0	0	
	Pupae Irradiated 32 Days Post-Larviposition (Group 3)													
Fertile Males	2	90.8	36.67 ± 2.11	44	82.50 ± 24.55	69.17 ± 28.46	116	5.50	13.90	16.70	33.30	30.60	2.64	78.50/64.80
40	4	83.2	71.67 ± 3.84	101	93.33 ± 0.00	82.50 ± 3.19	46	4.30	6.50	26.10	15.20	47.80	0.46	84.80/66.70
80	3	86.6	76.67 ± 3.84	77	89.17 ± 5.00	81.17 ± 5.77	8	12.50	37.50	37.50	12.50	0	0.10	(6/8)/(3/8)
100	3	91.3	60.00 ± 7.21	69	93.33 ± 0.00	84.17 ± 4.19	15	12.50	12.50	25	37.50	12.50	0.22	(11/15)/(7/15)
120	3	86.6	58.33 ± 7.53	73	98.33 ± 1.92	90.00 ± 5.44	0	0	0	0	0	0	0	
140	3	92.6	46.67 ± 6.32	62	94.16 ± 5.00	87.50 ± 7.39	2	0	0	0	0	100	0.03	(2/2)/(1/2)

**Table 2 insects-11-00522-t002:** Dissection results of *Glossina austeni* females mated with radiated male at different developmental stages.

Radiation Dose (Gy)	Insemination %	No. of Aborted Eggs	Uterus				
			Recently Ovulated Egg	Empty Due to Abortion	Viable Instar Larvae		
					I	II	III
Irradiated Adult Males							
Fertile males	93.90	40	33	16	2	3	8
40	100.00	201	0	71	0	2	12
80	93.30	209	1	74			
100	97.00	210	1	66			
120	100.00	224	7	80			
140	100.00	225	2	84			
Pupae Irradiated 36 Days Post-Larviposition (Group 1)							
Fertile males	100.00	45	32	3	11	8	6
40	100.00	261	37	45	2	4	5
80	99.00	268	17	84	0	0	0
100	100.00	280	23	67	0	0	0
120	100.00	241	21	62	0	0	0
140	100.00	288	14	82	0	0	0
Pupae Irradiated 34 Days Post-Larviposition (Group 2)							
Fertile males	100.00	40	17	12	1	4	9
40	100.00	121	10	45	0	3	4
80	100.00	156	12	54	1	0	0
100	100.00	151	9	51	0	0	0
120	100.00	167	10	68	0	0	0
140	100.00	178	7	93	0	0	0
Pupae Irradiated 32 Days Post-Larviposition (Group 3)							
Fertile males	100.0	15	24	0	1	3	7
40	98.80	190	14	60	1	3	6
80	98.50	185	17	49	1	1	0
100	100.00	175	21	40	0	0	0
120	100.00	182	17	46	1	0	0
140	100.00	135	5	53	0	0	0

**Table 3 insects-11-00522-t003:** Field cage evaluation of mating performance of *Glossina austeni* males irradiated with different irradiation doses and irradiated at different live stages (as adults or pupae).

Development Stage	Possible Pairs	Actual Mated	Overall Proportion (PM)	Relative Mating Index (RMI ± SD)	Mating Latency Time (Avg. ± SD)	Mating Duration (Avg. ± SD)	Mean Spermathecal Value	Insemination Rate
Adult Radiation	360	225	0.63	-	76.80 ± 0.04	149.20 ± 0.04	0.67 ± 0.15	0.99
Fertile Males	-	72	-	0.33 ± 0.13	81.00 ± 0.04	155.10 ± 0.04	0.64 ± 0.19	0.99
80 Gy	-	77	-	0.35 ± 0.16	79.00 ± 0.49	147.50 ± 0.05	0.72 ± 0.12	1.00
100 Gy	-	76	-	0.32 ± 0.12	69.90 ± 0.49	144.50 ± 0.05	0.65 ± 0.14	1.00
Adult vs. Pupae Radiation (100 Gy)	210	89	0.41	-	82.10 ± 0.04	117.70 ± 0.04	0.67 ± 0.26	0.97
Fertile Males	-	33	-	0.39 ± 0.02	85.20 ± 0.04	129.10 ± 0.04	0.70 ± 0.22	0.96
Adult	-	31	-	0.36 ± 0.14	71.70 ± 0.36	100.80 ± 0.03	0.58 ± 0.31	1.00
Pupae	-	21	-	0.24 ± 0.14	84.00 ± 0.06	114.90 ± 0.04	0.70 ± 0.24	1.00

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
