# Peer review of "Gamma Irradiation and Male Glossina austeni Mating Performance"

_insects, 2020, doi:10.3390/insects11080522_

Round 1

Reviewer 1 Report

Dear Authors, the research is interesting and well reported and described. There are only a few small tips (a few more comma, would make some sentences easier to follow).

Author Response

We are grateful for the effort of the reviewer to improve the quality of the manuscript.

All changes were made in the manuscript in track changes. Please see responses to the comments below.

Line 14: a comma was inserted after “component”

Line 15: a comma was inserted after “SIT”

Line 17: a comma was inserted after “radiation”

Line 30-31: “it was” was removed.

Line 31: “that” was removed.

Line 61: extra space before “Austria” was removed.

Line 111: “of G. austeni” was inserted after “pupae”

Line 113: “of G. austeni” was removed.

Line 231: The data was included into table 1

Line 253: a space was inserted before “and”

Reviewer 2 Report

I appreciate reading the paper but endured some difficulties in following the Results' section.

To make the text easier to follow I suggest to include sub-headings, such as: -effect of pupal age at irradiation on adult emergence timing; -effect of pupal age at irradiation on adult emergence rate; .......

Moreover, many of the numbers included into the text are difficult to follow. Consider moving to a table or to supporting files.

I did mark into the text some specific comments/suggestions.

Author Response

We are grateful for the effort of the reviewer to improve the quality of the manuscript.

All changes were made in the manuscript in track changes. Please see responses to the comments below.

Line 93: Inserted “Agricultural Research Council-Onderstepoort Veterinary Institute” before (ARC-OVI).

Line 110: the spelling of “Experimental” was corrected.

Line 112: “Agricultural Research Council-Onderstepoort Veterinary Institute” was moved to Line 93

Line 130: The full stop after “they” was removed.

Line 164: The following was inserted “(two treatment groups and one untreated group, 30 males per group)”

Line 175: “after three hours” was inserted after “mate”

Line 191: The statistical software name “GraphPadInstat” was inserted.

Line 206: Sub-headings were included in the results section.

Line 215: The emergence data was moved from the text to Table 1

Line 223: The data is now included in Table 1

Line 232: The text “that mated with irradiated males” was inserted.

Line 236: Many of the numbers have now been included in Table 1

Line 255: The text was changed to “both adult and pupae”

Line 265: The section of male survival was changed to indicate survival at day 30 this was also changed in the table. The linear correlation statistics were added.

Line 279: The first sentence of the 3.2 section was change to “Field cage experiments to assess mating performance of males irradiated with different irradiation doses and irradiated at different live stages (as adults or pupae) were carried out in September 2014 and November 2016, respectively.“Also a section was added after propensity of mating“ A propensity of mating (PM) (overall proportion of released females that mated)”. Two field change evaluations were done 1) mating performance of males irradiated with 80 and 100Gy.2) mating performance of males irradiated as adults and pupae. The propensity of mating is defined as the overall proportion of released females that had mated in each evaluation.

Line 313: Table 3 The title of the table was changed to “Field cage evaluation of mating performance of Glossina austeni males irradiated with different irradiation doses and irradiated at different live stages (as adults or pupae)”

Line 333: Sentence was changed to “In the present study a dose of 80 Gy or 100 Gy induced a minimum of 97% sterility in G. austeni females that mated with males treated either as adults or pupae. “

Line 341: Sentence was changed to “rate of pupae production”

Line 358: Reviewer 2 comment: this seems to be in contradiction with what stated on lines 354-355. Please explain. Both a negative and positive affect of radiation on longevity has previously been recorded for a range of insects not only tsetse. In the Caribbean fruit fly, it was shown that mild stress exposure early in life can have beneficial effects on performance later in life.

Line 389: Sentence was changed to “the assessment of longevity and matting success of 120 Gy-irradiated G. austeni males”

Line 394: the term "fitness" was replaced with “mating performance”

Line 429: Acknowledge person was included.
